# Metabolism as a New Avenue for Hepatocellular Carcinoma Therapy

**DOI:** 10.3390/ijms24043710

**Published:** 2023-02-13

**Authors:** Davide Gnocchi, Carlo Sabbà, Mara Massimi, Antonio Mazzocca

**Affiliations:** 1Interdisciplinary Department of Medicine, School of Medicine, University of Bari, Piazza G. Cesare 11, 70124 Bari, Italy; 2Department of Life, Health and Environmental Sciences, University of L’Aquila, 67100 L’Aquila, Italy

**Keywords:** hepatocellular carcinoma (HCC), metabolism, therapeutic approaches, multitarget metabolic systems

## Abstract

Hepatocellular carcinoma is today the sixth leading cause of cancer-related death worldwide, despite the decreased incidence of chronic hepatitis infections. This is due to the increased diffusion of metabolic diseases such as the metabolic syndrome, diabetes, obesity, and nonalcoholic steatohepatitis (NASH). The current protein kinase inhibitor therapies in HCC are very aggressive and not curative. From this perspective, a shift in strategy toward metabolic therapies may represent a promising option. Here, we review current knowledge on metabolic dysregulation in HCC and therapeutic approaches targeting metabolic pathways. We also propose a multi-target metabolic approach as a possible new option in HCC pharmacology.

## 1. Hepatocellular Carcinoma

The liver is a central organ of metabolic homeostasis as it is responsible for the detoxification process of xenobiotics, bile acid production, regulation of lipid metabolism, glycogen storage, amino acid biosynthesis, and nitrogen waste disposal through the urea cycle. In addition, the liver is involved in supporting immune function [1]. The predominant cell type in the liver parenchyma is the hepatocyte, which accounts for approximately 85% of the liver mass. Hepatocytes perform most of the metabolic functions of the liver.

Hepatocellular carcinoma (HCC), the most common primary cancer of the liver that originates in hepatocytes, is the sixth most commonly diagnosed neoplasm and the third leading cause of cancer-related death worldwide. From an epidemiological point of view, the incidence of HCC has decreased in some countries, such as China, Japan, and South Korea, while it has increased in Western countries [2,3]. Despite the reduced incidence of chronic hepatitis infections, recent projections estimate that HCC will rise to the third principal cause of death in Western countries by 2030 [4], and between 2020 and 2040, an increase of 55% is expected in the number of new HCC cases [2,3]. This alarming epidemiology can be explained by the increased incidence of metabolic diseases, such as the metabolic syndrome (MetS), diabetes, obesity, and nonalcoholic fatty liver disease (NAFLD) [5]. A direct link between the onset of NAFLD and the development of HCC has been reported [6]. Current therapeutic interventions for HCC include surgery, transarterial procedures (including TACE and TARE), and tyrosine kinase inhibitors (i.e., sorafenib, regorafenib, and lenvatinib), often in combination with immunotherapy (i.e., pembrolizumab and nivolumab) [5,7]. Despite these interventions, the prognosis of HCC patients is often poor, and pharmacological approaches are invasive and display many side effects. For this reason, a broader understanding of HCC pathogenesis is needed to develop efficacious and less aggressive pharmacological tools. To this end, the study of metabolism can provide concrete opportunities and provide interesting insights into this topic, as has happened in recent years.

## 2. Cancer and Metabolism

Genetics dominated cancer research for a long time after the pioneering work of Otto Warburg and Albert Szent-Gyorgyi in the 1930s [8,9,10,11,12]. Dysregulation of cellular metabolism has now reemerged as an important driver of cancer [13]. After decades of research primarily focused on tumor genetics, tumor cell metabolism has received renewed attention [13]. Currently, the scientific community is paying great attention to cellular metabolism and metabolic plasticity in tumor cell biology and has investigated the role of specific metabolic pathways in tumorigenesis [14,15]. Glycolysis, lactic acid fermentation, and oxidative phosphorylation (OXPHOS) are key metabolic processes that are now widely studied and considered targets for inhibiting tumor growth and overcoming drug resistance. In general, normal cells rely on the metabolic axis glucose→pyruvate→TCA/OXPHOS (oxidative glycolysis), whereas tumor cells shift towards lactic fermentation, glucose→pyruvate→lactate (fermentative glycolysis). This “fermentative signature” characterizes the Warburg effect, which is aerobic glycolysis or lactic acid fermentation regardless of the presence of hypoxic conditions [16]. Tumor cells primarily rely on the Warburg effect to generate energy, although this process is energetically inefficient (36 versus 4 ATP molecules/mol glucose) [17]. The switch to fermentative glycolysis provides several advantages to tumor cells, such as fast biosynthesis of ATP molecules, an increased supply of biosynthesis intermediates [17], and reduced generation of reactive oxygen species (ROS), which protects cancer cells from apoptosis [18]. Two steps are finely tuned: the conversion of pyruvate into lactate, which is catalyzed by lactate dehydrogenase (LDH), and the activation of OXPHOS. 

What triggers the Warburg effect is still being debated. Hypoxia, considered an activating stimulus, has recently been ruled out as a major factor, as some tumors develop the Warburg effect even under normoxic conditions [17]. It has been suggested that the Warburg phenotype is activated by an increased demand for NAD^+^ that is not matched with an adequate supply of ATP [19]. In general, it is now believed that the very early activation step of the Warburg effect could be an alteration of the metabolic control necessary for tumor cells to acquire a cell-autonomous nutrient uptake and an anabolic profile. Nevertheless, this initial trigger has not been fully elucidated. The mechanisms by which the Warburg effect supports drug resistance are also still under debate. It is known that the glycolytic environment can influence the response of cancer cells to drugs [20]. The relevance of the Warburg effect in tumorigenesis and drug resistance is still under debate [18,20], and the role of OXPHOS in cancer biology remains controversial. For example, some studies have shown that defective OXPHOS promotes tumorigenicity in hepatocellular carcinoma (HCC) [21,22], while others have reported inhibition of OXPHOS as a therapeutic strategy for HCC [23,24]. Similar differences have been described in other types of malignancies [25,26].

## 3. Hepatocellular Carcinoma and Metabolism

Metabolic dysregulation is emerging as an important risk factor for HCC [6]. As in other tumors, the neoplastic transformation of a normal hepatocyte is characterized by a general metabolic rewiring. Transformed hepatocytes display the Warburg phenotype, which in the liver is characterized by rearrangements in the expression and phosphorylation of key glycolytic enzymes [27]. Indeed, more than fifty years ago, metabolic reprogramming to enhance glycolysis was shown to be central to transforming hepatocytes [28]. Glucose metabolism rewiring can find a mechanistic explanation in the hypoxic milieu in which HCC lies. In fact, in normal hepatic tissue, the median O_2_ partial pressure is about 30 mmHg, whereas in the HCC intratumor region, it is roughly 6 mmHg [29,30,31]. In line with this, HIF-1α is highly expressed in HCC [32]. In this hypoxic microenvironment, LDH activity is increased [33] and the pyruvate dehydrogenase complex (PDH) is reduced [34], thus obtaining an increase in lactate production and a dampening of the conversion of pyruvate into acetyl-CoA, which contribute to the instauration of a glycolytic phenotype.

### 3.1. Glucose Metabolism Rewiring

Glucose homeostasis is rearranged in HCC, leading to a change in the expression and activity of transporters and enzymes, which in turn leads to a rewiring of the metabolite flux. The Warburg phenotype in HCC is not characterized by increased gluconeogenesis because fructose 1,6-bisphosphatase 1 (FBP1) and phosphoenolpyruvate carboxykinase 1 and 2 (PEPCK 1 and 2) expression are reduced [35]. In addition, the expression of glucose transporters GLUT1 and GLUT2 is increased in HCC, and while GLUT2 overexpression worsens the prognosis of patients, the inhibition of GLUT1 expression attenuates the malignant behavior of HCC cells [36,37,38]. The conversion of glucose into glucose-6-phosphate (G6Ph) is a highly regulated metabolic reaction catalyzed by the hexokinase (HK) enzyme, which has five isoforms: HK1 to HK4 and the hexokinase domain containing 1 (HKDC1). The expression of HK2, the prevalent isoform in HCC, is associated with reduced survival in HCC patients [39], and, accordingly, HK2 inhibition dampens glycolysis in favor of OXPHOS, relieving the tumor phenotype [40]. HKDC1 expression has also been shown to exacerbate HCC aggressiveness and prognosis [41].

The glycolytic phenotype is strengthened by the modification of the expression and activity of other enzymes. For example, glyceraldehyde-3-phosphate dehydrogenase (GAPDH) [42,43] and pyruvate kinase 2 (PKM2) [44] are upregulated in HCC.

HCC also exhibits an increased dependence on glutamine metabolism [45,46]. Glutamine undergoes different metabolic routes in HCC. For example, glutaminolysis leads to α-ketoglutarate (α-KG) production, which supports the TCA cycle for energy production in the context of low-efficiency OXPHOS. Moreover, α-KG can be converted into citrate, thus fueling the biosynthesis of lipids. However, there is no consensus regarding the function of glutamine in HCC. Some authors proposed a general glutaminolysis fate with a negligible contribution to reductive carboxylation, which leads to lactate production [40]. This notion was not supported by other authors who found inhibition of the pathway leading to α-KG production [35], possibly reflecting the intrinsic heterogeneity of tumor tissue [47]. LDH is a key enzyme in metabolic reprogramming in many tumors and HCC. Five isoforms have been described in humans, and LDHA and LDHB are the most expressed isoforms in the liver. In HCC, serum LDH is a reliable tumor biomarker [48,49]. Lactate is central to metabolic rearrangements in most tumors and HCC, and elevated serum levels have been observed in HCC patients [50]. In HCC, part of lactate is secreted in the extracellular medium by monocarboxylate transporters (MCT), in particular by MCT4, whose expression is correlated with proliferation through Ki-67 [51]. This process is regulated by the transmembrane protein CD147, which elicits MCT1 expression, which in turn increases lactate externalization [52]. MCT1 expression, just like MCT4, promotes HCC growth [52]. Higher plasma levels of lactate have been detected in patients with NASH compared with healthy subjects, supporting the potential role of lactate as an active metabolic mediator of the disease [53].

### 3.2. Pentose Phosphate Pathway

The Pentose Phosphate Pathway (PPP) plays a central role in the metabolic reprogramming of tumor cells and HCC. The main outcome of the PPP is the production of ribose-5-phosphate, which is the precursor for the synthesis of nucleotides that are needed for tumor cell growth. In addition, NADPH is another product of the PPP that sustains lipid biosynthesis and keeps the tumor cell in an antioxidant environment by maintaining thioredoxin and glutathione in a reduced state. This may help tumor cells protect themselves from apoptotic cell death and develop resistance to drug therapy [54]. Although the levels of ribulose 5-phosphate (Ru5P) and ribose 5-phosphate (R5P) are decreased due to their rapid utilization by cancer cells, the transcription of most enzymes involved in the PPP is elicited in HCC [55]. Among the enzymes that are involved in PPP rewiring, glucose-6-Ph dehydrogenase (G6PDH) should be mentioned. The expression of this enzyme was shown to result in increased metastasis, drug resistance, and decreased survival in HCC. Notably, HBV protein X (HBx) triggers G6PDH expression through a Nrf2-mediated mechanism [56]. Furthermore, in non-oxidative PPP, Nrf-2 upregulates transketolase (TKT) by reducing oxidative stress and increasing purine biosynthesis during hepatocarcinogenesis [57].

### 3.3. TCA Cycle Rewiring

The TCA cycle produces NADH and FADH_2_, which fuel OXPHOS activity. The TCA cycle is dysregulated at several points in HCC. Pyruvate dehydrogenase kinase 4 (PDK4), which phosphorylates and inhibits pyruvate dehydrogenase (PDH), is downregulated in HCC and associated with increased lipid biosynthesis and poor prognosis [58]. Succinate dehydrogenase (SDH) [succinate→fumarate] and fumarate hydratase (FH) [fumarate→malate] are deficient in HCC. Moreover, succinate and fumarate-mediated stabilization of HIF-α enhances glycolysis and angiogenesis [59]. Interestingly, SDHB deficiency promotes the malignancy of HCC, triggering the Warburg effect [60].

### 3.4. OXPHOS Rewiring

In normal cells, OXPHOS utilizes NADH and FADH_2_ from the TCA cycle for ATP production. Several pieces of evidence associate defective OXPHOS with enhanced HCC development through different mechanisms, for example, mitoribosome defects or by TGFβ-mediated reduction of OXPHOS without the involvement of the glycolytic pathway [21,22]. HCV-mediated reduced expression of OXPHOS protein subunits leads to a Warburg-like phenotype, eventually promoting HCC development [61]. Membrane trafficking-involved Rab3A GTPase is overexpressed and also post-translationally modified by the addition of N-acetylglucosamine (O-GlcNAcylated) in HCC. Usually, unmodified Rab3 inhibits the metastatic process in HCC by boosting OXPHOS, but O-GlcNAcylation impedes this function [62]. Notably, mitochondrial miR-181a-5p is upregulated in HCC, and this results in a downregulation of cytochrome B and cytochrome C oxidase 2 (COX2) with consequent impairment of the electron transport chain. Furthermore, glucose metabolism is reprogrammed by increasing the expression of the glucose transporters GLUT1 and hexokinase 2 (HK2), leading to high glucose uptake and upregulation of LDH activity [63]. Interestingly, the correlation between cell stemness, metabolism, and HCC is provided by the homeobox transcription factor NANOG2, whose expression is increased in alcohol- and obesity/HCV-induced HCC mouse models as well as in human tumor-initiating cells, where NANOG inhibits OXPHOS [64].

### 3.5. Lipid Metabolism Rewiring

Lipid metabolism is one of the important metabolic tasks accomplished by the liver. Altered lipid metabolism homeostasis is responsible for many pathologic conditions. Dyslipidemias may lead to diabetes, obesity, and liver steatosis, and all these conditions have been correlated with HCC onset [65,66]. De novo lipogenesis (DNL) is very well regulated, and several hindrances have been reported for DNL in HCC. In general, DNL follows this route: pyruvate→acetyl-CoA+oxaloacetate (OAA)→citrate [inside mitochondria]. In conditions of carbohydrate abundance, citrate is exported into the cytoplasm where ATP-citrate lyase (ACLY) converts citrate into acetyl-CoA+OAA. Then, lipid biosynthesis takes place in the cytoplasm, mainly by acetyl-CoA carboxylase (ACC) [acetyl-CoA→malonyl-CoA] and fatty acid synthase (FASN) [malonyl-CoA→palmitoyl-CoA] [67]. DNL works in close association with the PPP to produce the necessary NADPH. DNL deregulations have been reported in HCC-predisposing conditions, such as NAFLD [68]. In addition, increased expression of several DNL enzymes as well as enzymes involved in NADPH biosynthesis, such as G6PD and malic enzyme, has been reported [35]. FASN overexpression is associated with increased cancer growth and survival and a worse prognosis in patients [68,69]. The acetyl-CoA can also derive from acetate, which is imported into cells by MCTs transporters. Acetyl-CoA synthetase enzymes (ACSSs) then convert acetate into acetyl-CoA. ACSS2 is present in the cytosol, while ACSS1 and ACSS3 are mitochondrial isoforms. Notably, the expression of ACSS1 was increased and correlated with enhanced malignant features in HCC [35].

HCC uses DNL and exogenous fatty acids to meet its growth requirements [70], and increased free fatty acid uptake through the fatty acid translocase CD36 correlates with the initiation and progression of HCC [71].

The liver β-oxidizes fatty acids in either peroxisomes or mitochondria, where they are transported by carnitine palmitoyltransferase-I (CPT-I), which condenses carnitine with very long Acyl-CoA [72]. Interestingly, NASH-induced rats show reduced CPT-I activity as well as defective complexes I and II of the mitochondrial electron transport chain (ETC) [73,74]. Furthermore, HCC patients showed defective mitochondrial β-oxidation due to decreased expression of several enzymes involved [35]. Thus, impaired mitochondrial functions can be considered predisposing conditions for HCC development.

Cholesterol homeostasis is another important determinant of HCC development. The key liver enzyme that controls cholesterol biosynthesis is 3-hydroxy-3-methylglutaryl-CoA reductase (HMG-CoA reductase), whose reaction product is mevalonate. Some reports have described an association between HMG-CoA reductase/mevalonate and HCC. In fact, from a clinical point of view, several reports described statins as dampening the probability of developing HCC. Corroborating observations come from in vitro and mouse studies [75,76,77,78,79]. The pro-HCC effects of bile acids, cholic acid, and deoxycholic acid have also been described [80,81]. Phospholipid homeostasis is also involved in HCC pathogenesis. Cirrhotic rats showed altered activity of phospholipid biosynthesis enzymes [82]. Interestingly, HCC patients have high serum and tissue levels of phosphatidylcholine (PC) [83].

### 3.6. Nucleotide Metabolism Rewiring

Nucleotide metabolism is essential for nucleic acid biosynthesis, and this is particularly true for tumor cells that have an increased proliferative rate compared to healthy cells. In this regard, the PPP provides R5Ph, which is needed along with amino acids for nucleic acid biosynthesis. In HCC, the expression of several genes involved in this biosynthetic pathway is increased. In particular, three enzymes constitute the enzyme complex that catalyzes the initial reaction in pyrimidine biosynthesis: carbamoyl phosphate synthase 2, aspartate transcarbamylase, and dihydroorotase [84,85]. Furthermore, other rate-limiting enzymes, such as thymidylate synthase (TYMS), thymidine kinase 1 (TK1), and deoxythymidylate kinase (DTYMK), are upregulated in HCC and are associated with poor prognosis and cancer stemness [86,87,88]. The expression of most of the enzymes involved in purine biosynthesis is also increased in HCC [55,87,88].

### 3.7. Protein and Amino Acid Metabolism Rewiring

The liver is also a central hub for protein and amino acid homeostasis, so this imbalance due to pathological conditions reflects changes in protein and amino acid levels. For example, negative correlations between serum albumin levels and tumor diameter, tumor multifocality, portal vein thrombosis, and α-fetoprotein, have been reported [89]. The liver is also responsible for the biosynthesis of non-essential amino acids (NEAAs), which are involved in many biochemical pathways. In many tumors, NEAA production is increased due to the rewiring of glucose metabolism, which increases the demand for oxidizable substrates [90,91,92,93]. For example, in HCC patients, the levels of most NEAAs were significantly higher compared to healthy subjects, while branched-chain amino acids (BCAAs) were decreased compared to aromatic amino acids [94,95]. Interestingly, NEAA actively supports HCC by mediating adaptation to hypoxia [96]. It is worth mentioning here the behavior of three amino acids: glutamine, aspartic acid, and glycine. In HCC, glutamine and aspartate levels are elevated, and glycine levels are decreased [83]. 

### 3.8. Urea Cycle Rewiring

In HCC, most urea cycle enzymes are downregulated, along with low concentrations of urea cycle metabolites such as arginine, ornithine, and citrulline, which are common in advanced HCC stages. Two key enzymes in the urea cycle, argininosuccinate synthase 1 (ASS1) and carbamoyl phosphate synthase (CPS), show a hypermethylated state in HCC, which explains their reduced expression.

### 3.9. HCC Tumor Microenvironment Rewiring

Hepatocytes are the main cell type in the liver parenchyma, while the rest is represented by stromal cells, including endothelial cells, immune cells, Kupffer cells, and hepatic stellate cells. Stromal cells are also recruited by HCC cells in the tumor microenvironment (TME). Two important effector cells are tumor-associated macrophages (TAM) and myeloid-derived suppressor cells (MDSC), which are the main cell types found in the tumor infiltrate [97]. The crosstalk between cancer cell-released metabolites and immune cells sustains the tumor ecosystem. Indeed, upon an inflammatory challenge, macrophages acquire an inflammatory cytokine-producing M1-like phenotype, whereas anti-inflammatory stimuli lead to an immunosuppressive M2-like phenotype [98]. Hence, tumor onset is favored by macrophage-induced chronic inflammation. The immunosuppressive M2-like phenotype may then prevail, supporting growth, angiogenesis, and the metastatic process [99]. These observations are in line with the consolidated view that an inflammatory status promotes tumor onset. For example, tumor-secreted cytokines attract monocyte-derived TAM, which switches to a glycolytic fermentative metabolism and survives in the hypoxic TME [99,100]. Notably, M2-like TAMs may be involved in mediating drug resistance and promoting cancer stemness in HCC, resulting in a poorer prognosis for HCC patients [101,102]. Secreted lactate mediates the crosstalk between TAM and cancer cells. This promotes a VEGF-induced transition to an M2-like phenotype and inhibition of NK cells, by favoring MDSC [103,104]. MDSC-mediated immunosuppression can also be boosted by fatty acids, whose uptake is mediated by CD36 [105].

## 4. Metabolism-Based Pharmacology Approaches in HCC

Several pharmacological approaches have been and are being developed using the HCC metabolic features described above. Several chemicals are currently being investigated, some in their initial preclinical stages and others in clinical trials.

### 4.1. Inhibition of the Glycolytic Pathway and Pentose Phosphate Pathway

Several glycolysis inhibitors are currently being investigated as possible therapeutic molecules. The glucose analog 2-deoxy-D-glucose (2-DG) is converted to 2-DG-6-phosphate, which inhibits HK2 activity in a non-competitive manner. This inhibition is associated with reduced HCC growth and invasiveness. Moreover, 2-DG synergizes with the effect of sorafenib in HCC cell lines [106,107]. 3-Bromopyruvic acid (3-BrPA) inhibits several enzymes, including HK2, PGK1, succinate dehydrogenase, GAPDH, and LDH. Also, some studies have identified GAPDH as the principal target of 3-BrPA in HCC cell lines [43,108,109,110]. The plant-derived molecule koniginic acid also inhibits GAPDH by binding to the enzyme’s active site. However, the potential applicability of koniginic acid in HCC has not been well demonstrated [111]. Several metabolites from plants are considered potential inhibitors of glycolysis in HCC. Among them, chrysin, a flavonoid found in Chinese medicinal plants, can reduce the expression of HK2, reduce glucose uptake and lactate production, and ultimately trigger the mitochondrial apoptotic pathway. The anti-HCC effects of chrysin have also been described in a mouse model [112]. Likewise, the plant-stress metabolite methyl jasmonate reduces mitochondrial transmembrane potential by sequestering HK2 and thereby elicits apoptosis in HCC cell lines and mice [113]. The naphthoquinone shikonin from *Lithospermum erythrorhizon* displays anti-HCC effects and enhances the effect of sorafenib [114]. The plant-derived isoflavone genistein inhibits glycolysis by targeting HIFα, HK2, and GLUT1. In addition, genistein enhances in vitro and in vivo sensitivity to sorafenib [115]. Finally, the anthraquinone emodin, present in different plant species, inhibits glycolysis by reducing HK2, PKM2, and LDHA expression [116]. The thiamine antagonist oxythiamine, which inhibits transketolase (TKT), inhibits HCC cell growth and synergizes with sorafenib by eliciting ROS production in vitro and in vivo [57].

### 4.2. Lipid Metabolism Rewiring Inhibition

To date, little evidence exists to suggest the possibility of lipid metabolism as an effective therapeutic target in HCC. For example, the inhibition of the central lipogenic enzyme acetyl-CoA carboxylase (ACC) has been considered for pharmacological purposes. Indeed, some chemical inhibitors of ACC are now under investigation, such as ND-654, an allosteric inhibitor of ACC1 and ACC2, which also enhances the pharmacological effects of sorafenib [117]. Several fatty acid synthase (FASN) inhibitors, including orlistat, C93, C75, GSK2194069, and GSK837149A, have been evaluated in preclinical testing [118,119]. These compounds, however, show several side effects and significant toxicity. One stearoyl-CoA desaturase 1 (SCD1) inhibitor, A939572, was shown to be efficacious in HCC cell lines and to increase sorafenib sensitivity by modulating ER stress-induced differentiation [120]. The transcription factor Liver X receptor alpha (LXRα) is involved in lipid homeostasis and is under evaluation as an anti-HCC target. TO901317 has been reported to increase LXRα expression and decrease glucose uptake by inhibiting GLUT1 expression [121]. Finally, the squalene synthase (SQS) enzyme in the cholesterol biosynthetic pathway has been considered a druggable target for HCC. For example, YM-53601 was reported to inhibit SQS in vivo and synergize with doxorubicin to inhibit HCC growth [122]. Moreover, an ACLY inhibitor was recently shown to induce a pro-apoptotic effect in in vitro models of HCC [123]. No evidence is available so far about the potential toxicity of these last-mentioned compounds.

Of note, an important aspect to consider in this context is the role of circadian rhythms in the regulation of lipid metabolism and, hence, in controlling the development of predisposing dysmetabolic conditions leading to HCC, such as NAFLD. Indeed, dysregulation of the circadian clock is emerging as an important aspect in the pathogenesis of several diseases [124,125]. In particular, the role of NAFLD in the development of HCC has been recognized [126]. Thus, strategies aimed at targeting the circadian clock machinery can be considered as a possible option in the management of HCC and predisposing dysmetabolic conditions.

### 4.3. Other Potential Metabolic Targets for HCC

So far, chemicals targeting other metabolic pathways have not provided conclusive indications. For example, it has been proposed to target epigenetic regulation and ubiquitin-dependent degradation to increase fructose-1,6-bisphosphatase 1 (FBP1) expression, which is decreased in HCC [127]. Several molecules are under testing in this respect, such as histone deacetylase inhibitors or lysine-specific histone demethylase 1 inhibitors [128,129,130]. Still, these chemicals show low specificity and a pleiotropic action. Proteasome inhibitors and dexamethasone have also been suggested to regain FBP1 levels, but likewise, the scarce selectivity results in many side effects [131]. Glutaminase 1 (GLS1), which converts glutamine into glutamate, is a key enzyme in tumor cell metabolism because it can promote lipid and nucleotide biosynthesis and reduce ROS generation, thereby protecting tumor cells from apoptosis. Some GLS1 inhibitors have been tested, but they showed toxicity and low selectivity [132,133]. Chemical inhibitors of glutamine uptake are also under evaluation [134,135]. Among them, the plant metabolite berberine shows interesting effects in vitro [136]. Some chemicals targeting metabolism are now under clinical trial evaluation, but the results so far are not conclusive.

In this scenario, it is important to mention that there is growing interest in the potential use of AMP-activated protein kinase (AMPK) activators to improve metabolic abnormalities and better manage neoplastic conditions, including HCC [137]. In fact, some AMPK activators are now being considered as possible drugs for the treatment of metabolic disorders and cancer [138]. In HCC, a recent report suggested that AMPK activation shows therapeutic potential [139]. Still, caution is suggested since contrasting results have been reported for cancer therapy [140]. Interestingly, AMPK activation was shown to prevent thrombotic conditions by phosphorylation and consequent inactivation of ACC [141]. This can represent an important aspect since a prothrombotic imbalance was observed in NASH/cirrhosis, which predicts the development of HCC [142].

An overview of the metabolic targets in HCC, for which chemical inhibitors are available, is shown in Figure 1.

## 5. Multitarget Metabolic Systems

### 5.1. Plant-Based Multi-Pathway Approach

As mentioned above, metabolism-based single-target approaches still show several critical points in terms of efficacy and side effects. To this end, a synergistic approach targeting multiple metabolic pathways or processes may offer some advantages to achieve higher efficacy, a lower dose, and lower toxicity. For example, plant extracts, because their composition is a blend of several metabolites, can provide an interesting option to target several metabolic pathways at the same time. Plant extracts have not been explored much in this direction. However, several studies have reported interesting effects of plant extracts on several types of tumors, most likely due to synergistic effects targeting multiple pathways [143,144,145,146,147]. The plant-derived polyphenol molecule resveratrol acts on multiple cellular targets and exhibits a wide range of biological activities, including positive effects on various metabolic pathways [148]. For example, resveratrol reduces the utilization of glucose and amino acids for energy production and the release of lactic acid in HCC cells [149]. Furthermore, resveratrol-derived natural metabolites have multiple effects on glucose metabolism in hepatoma cells [150]. Recently, we investigated the role of *Crithmum maritimum*, an edible wild plant that grows spontaneously along the Mediterranean and Atlantic coasts. We performed an extensive screen for the chemical composition and biological effects of *Crithmum maritimum* harvested along the Apulian coast. Notably, we found a significant effect of an ethyl acetate extract of *Crithmum maritimum* in inhibiting HCC cell growth by exerting synergistic effects on several metabolic pathways involved in promoting HCC, such as glycolysis/Warburg effect, lipid and cholesterol metabolism, and amino acid metabolism. We also identified the composition of the extract, which contains high concentrations of falcarindiol and chlorogenic acid, both of which have anti-cancer properties [151]. Interestingly, *Crithmum maritimum* modulates the bioenergetic profile of HCC cells by increasing OXPHOS and decreasing LDH activity, and hence lactate production [152]. Moreover, *Crithmum maritimum* increases the expression of markers typical of healthy hepatocytes [153]. Of note, *Crithmum maritimum* can also be a valuable option to improve conventional pharmacological treatments with sorafenib [154]. Indeed, we found that *Crithmum maritimum* can be used in association with sorafenib to decrease the dose and the toxicity. This is an interesting example of how a multi-targeting approach in metabolism can improve conventional pharmacological cytotoxic approaches in HCC. As far as we know, this approach has not been much investigated, and more research is needed to fully uncover its actual potential.

### 5.2. Targets Exploiting Synergistic Effects on Metabolic Pathways

In addition, and in support of a plant-based multi-pathway approach, selective inhibition of receptors controlling metabolism may offer an additional translational opportunity for HCC pharmacological management. In this respect, we have recently demonstrated that LPA receptor 6 (LPAR6) is directly involved in the control of HCC cell metabolism [155] and that ectopic expression of LPAR6 in HCC cells drives sorafenib resistance by triggering a “metabolic switch”, which increases lactic acid fermentation at the expense of OXPHOS. This sorafenib resistance can be overcome by reducing lactic acid fermentation through inhibition of LPAR6 using our recently developed novel LPAR6 antagonists [156,157,158]. LPAR6 has a pro-tumorigenic role in HCC [159] by controlling the trans-differentiation of peritumoral tissue fibroblasts (PTFs) into carcinoma-associated fibroblasts (CAFs) [160] and its overexpression leads to a worse clinical outcome in HCC patients [159]. In this context, we emphasize the importance of the balance between fermentative glycolysis/OXPHOS in tumor development and cancer drug resistance [161].

Thus, metabolism-based interventions should be considered a valid option to support current therapies in patients with HCC. Current and proposed metabolism-based interventions are shown in Figure 2.

## 6. Conclusions and Future Perspectives

After several decades in which cancer research has been focused on tumor genetics and molecular biology, a new interest has arisen in the role of metabolism in driving and sustaining carcinogenesis. As discussed above, many biological pathways have been extensively studied, and several enzymes have been identified as potential therapeutic targets. Still, the complex intersections between the different pathways require a more structured, multitarget, process-directed approach aimed at targeting more processes at the same time to minimize possible “escape” strategies of the tumor. The biological mechanisms by which normal cells become transformed and acquire a tumor phenotype are still not fully elucidated. This is an essential step for the basic knowledge of how cells work and how they start to not work properly. From this perspective, a deeper understanding of how cellular adaptation is driven by metabolic processes is central. 

HCC is becoming a serious health issue, especially in Western countries, and there is an extreme need for novel and effective therapeutic approaches. Many biochemical pathways are dysregulated in HCC, and a richer knowledge of these fields has the concrete potential to disclose new perspectives for HCC prevention and therapy. In this context, targeting single enzymes involved in the regulation of metabolic pathways can be an option. Still, so far, the results obtained are not conclusive in terms of efficacy and tolerability. This may be due to the intrinsic pleiotropic capacity of tumor cells to bypass a single inhibitory step. For this reason, a synergistic multitarget/multi-process approach can prove useful. In this respect, plant extracts may represent a valuable tool due to their blended composition, which can target many processes at the same time. Coupled with this approach, selective targeting of specific metabolism-controlling receptors may offer a supplemental option, as described above for LPAR6. 

In conclusion, alterations in cellular metabolism are important drivers of the tumorigenesis process and the promotion of drug resistance [162,163,164,165]. In this context, the balance between glycolysis and OXPHOS is a central point, and the mechanisms by which this is regulated are still not fully understood. We believe that the complex interplay between tumor cell metabolism and drug resistance is an important aspect, and that the balance between OXPHOS and fermentative glycolysis is critical for determining the cellular behavior during neoplastic transformation and the acquisition of drug resistance by tumor cells. Furthermore, it is worth mentioning that, in our opinion, modulating the balance between OXPHOS and lactic acid fermentation to trigger OXPHOS and inhibit fermentation could be considered a strategy to reduce tumor invasiveness and drug resistance. Indeed, we recently demonstrated that inhibition of OXPHOS in favor of lactic acid fermentation elicits drug resistance in HCC [156]. Based on this observation, approaches to modulate the metabolism of fermenting tumor cells may provide tools to overcome cancer drug resistance. In particular, we propose the activation of OXPHOS and the inhibition of lactic acid fermentation/Warburg phenotype to reduce tumor growth and drug resistance, as well as a pro-differentiation strategy. In line with this principle, this “metabolic approach” could be independent of the genetic background of each tumor, and this could be exploitable for therapeutic purposes in different types of malignancies.

## Figures and Tables

**Figure 1 ijms-24-03710-f001:**
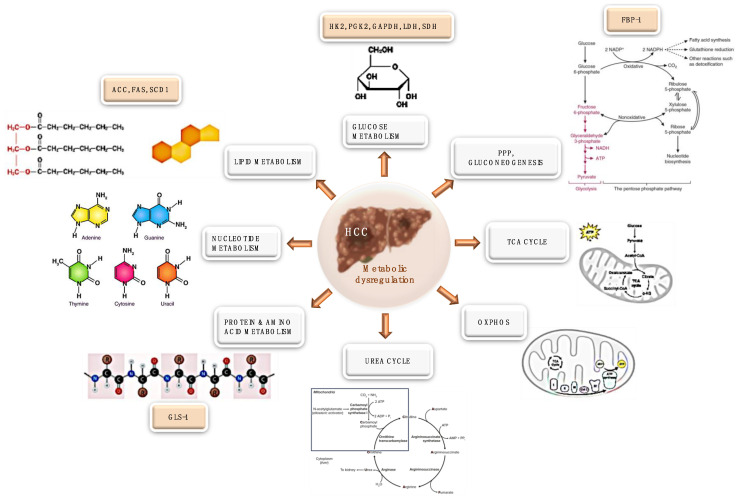
Metabolic enzymes that can be targeted in HCC. The metabolic pathways are depicted in white boxes, while metabolic targets are reported in light brown boxes. See the text for details. Illustrations have been realized with Biorender.

**Figure 2 ijms-24-03710-f002:**
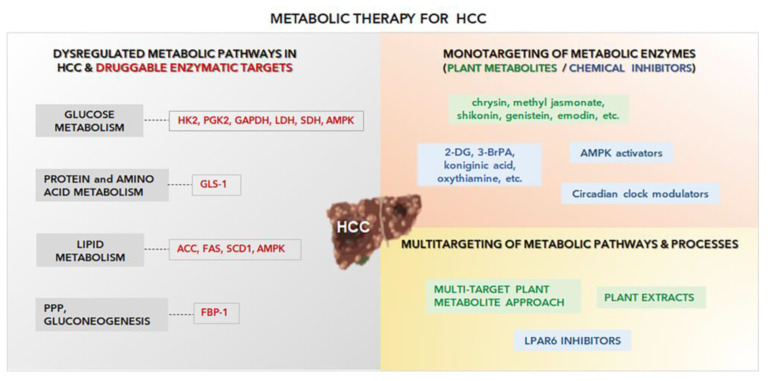
Current therapies and metabolism-based interventions for HCC. New metabolism-based therapeutics may provide promising tools for the treatment of HCC. See the text for details. Illustrations have been realized with Biorender.

## Data Availability

Not applicable.

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
