# Peer review of "Metabolism as a New Avenue for Hepatocellular Carcinoma Therapy"

_ijms, 2023, doi:10.3390/ijms24043710_

Round 1

Reviewer 1 Report

In the present review the authors provide an interesting and in-depth overview of multiple metabolic changes that are critically involved in the HCC pathogenesis. Also, this work outlines the current state of art of the new  strategies proposed to target key hubs of metabolic pathways, as an alternative to standard pharmacology. In my opinion, this report adds an important contribution to literature regarding potential new therapeutic approaches for the treatment of HCC. 

I would suggest just a few minor corrections/revisions (eg line 196: the acronym PPP is introduced here for the first time)

Author Response

Response to Reviewer 1 Comments

In the present review the authors provide an interesting and in-depth overview of multiple metabolic changes that are critically involved in the HCC pathogenesis. Also, this work outlines the current state of art of the new strategies proposed to target key hubs of metabolic pathways, as an alternative to standard pharmacology. In my opinion, this report adds an important contribution to literature regarding potential new therapeutic approaches for the treatment of HCC. 

  1. I would suggest just a few minor corrections/revisions (eg line 196: the acronym PPP is introduced here for the first time).

Response: We thank the reviewer #1 for his/her positive comments and for pointing out this oversight. The acronym is specified for the first time in the section “3.2 Pentose Phosphate Pathway” and then repeated in sections 3.5, 3.6 and 4.1. In section 4.1 we have avoided the acronym in the title for clarity.

Reviewer 2 Report

Overall, this review manuscript has nicely reviewed important aspects of metabolism associated avenues for HCC. 

Some suggestions for authors as follows

- a comprehensive graphical abstract diagram is suggested. it will help audience engage better with the manuscript and will enhance its readibility. 

- Figures 1 and 2 are very basic diagrams and authors are suggested to upscale these diagrams using more data and publishable graphics. 

Author Response

Response to Reviewer 2 Comments 

Overall, this review manuscript has nicely reviewed important aspects of metabolism associated avenues for HCC.

Some suggestions for authors as follows:

  1. A comprehensive graphical abstract diagram is suggested. it will help audience engage better with the manuscript and will enhance its readability.

Response: We thank the Reviewer #2 for his/her positive comments and helpful suggestions. We have now changed the graphical abstract as to make it more informative and comprehensive.

  1. Figures 1 and 2 are very basic diagrams and authors are suggested to upscale these diagrams using more data and publishable graphics.

Response: We improved and added more details to Figure 1 and 2 as suggested.  

Reviewer 3 Report

The authors provide an overview of the growing understanding of the role of metabolism in HCC.  The paper reads well and will likely be of interest to some working in this area.  A few minor comments are provided below:

1.  There are several errors in the section on lipid metabolism.   For example when describing the DNL route the authors are encouraged to review articles in this area such as (PMID: 35031766) as the route is not pyruvate-acetyl-CoA-aceytyl-CoA+oAA-citrate.  Acetyol-CoA goes to citrate in the TCA and then Citrate is exported out of the mitochondria and then convereted to acetylyl-CoA+oAA by ACLY not the other way around as currently written.

2,  With respect to metabolic therapies and HCC the authors seem to have missed an important target that is regulated by both small molecules and xenobiotics and that is AMPK.  Several papers in this area demonstrate that AMPK reduces fibrosis, inflammation, lipogenesis and enhances fatty acid oxidation and tumor immunogenicity as recently reviewed (PMID: 36316383)

3.  More discussion about the mechanisms of how metabolic targets enhance the effects of cytotoxic therapies is needed as currently this section while a major focus of the review is quite vague.  If not known authors should address what the key questions are that need to be answered.

Author Response

Response to Reviewer 3 Comments 

The authors provide an overview of the growing understanding of the role of metabolism in HCC. The paper reads well and will likely be of interest to some working in this area. A few minor comments are provided below:

  1. There are several errors in the section on lipid metabolism. For example, when describing the DNL route the authors are encouraged to review articles in this area such as (PMID: 35031766) as the route is not pyruvate-acetyl-CoA-aceytyl- CoA+oAA-citrate. Acetyol-CoA goes to citrate in the TCA and then Citrate is exported out of the mitochondria and then convereted to acetylyl-CoA+oAA by ACLY not the other way around as currently written.

Response: We thank the Reviewer #3 for his/her comments and helpful suggestions. We have now amended the section on lipid metabolism accordingly and added the suggested reference (pag. 5, lines 204-209).

  1. With respect to metabolic therapies and HCC the authors seem to have missed an important target that is regulated by both small molecules and xenobiotics and that is AMPK. Several papers in this area demonstrate that AMPK reduces fibrosis, inflammation, lipogenesis and enhances fatty acid oxidation and tumor immunogenicity as recently reviewed (PMID: 36316383).

Response: We thank the Reviewer #3 for this important suggestion. We have now added a paragraph on the role of AMPK in HCC metabolism and included the suggested reference (pag. 8, lines 376-386).

  1. More discussion about the mechanisms of how metabolic targets enhance the effects of cytotoxic therapies is needed as currently this section while a major focus of the review is quite vague. If not known authors should address what the key questions are that need to be answered.

Response: We thank the Reviewer #3 for this insightful observation. We have now included a paragraph in the section “5.1 Plant-based multi-pathway approach”, in which we discuss how metabolic targets can enhance current cytotoxic therapies. Indeed, so far not much is known on this topic and further research is needed to characterize this interesting field (pag.10, lines 425-431).